# Diversity in Proprotein Convertase Reactivity among Human Papillomavirus Types

**DOI:** 10.3390/v16010039

**Published:** 2023-12-26

**Authors:** Gonzalo Izaguirre, Lam Minh Uyen Phan, Shaan Asif, Samina Alam, Craig Meyers, Lijun Rong

**Affiliations:** 1Department of Periodontics, College of Dentistry, University of Illinois Chicago, Chicago, IL 60612, USA; 2Departments of Microbiology and Immunology, College of Medicine, Penn State University, Hershey, PA 17033, USA; 3Departments of Microbiology and Immunology, College of Medicine, University of Illinois Chicago, Chicago, IL 60612, USA

**Keywords:** host proteases, virus activation, virus infectivity, human papillomavirus, proprotein convertases, furin, protease inhibitors, antivirals

## Abstract

The cleavage of viral surface proteins by furin is associated with some viruses’ high virulence and infectivity. The human papillomavirus (HPV) requires the proteolytic processing of its capsid proteins for activation before entry. Variability in reactivity with furin and other proprotein convertases (PCs) among HPV types was investigated. HPV16, the most prevalent and carcinogenic HPV type, reacted with PCs with the broadest selectivity compared to other types in reactions of pseudoviral particles with the recombinant PCs, furin, PC4, PC5, PACE4, and PC7. Proteolytic preactivation was assessed using a well-established entry assay into PC-inhibited cells based on the green fluorescent protein as a reporter. The inhibition of the target cell PC activity with serpin-based PC-selective inhibitors also showed a diversity of PC selectivity among HPV types. HPV16 reacted with furin at the highest rate compared to the other types in time-dependent preactivation reactions and produced the highest entry values standardized to pseudoviral particle concentration. The predominant expression of furin in keratinocytes and the high reactivity of HPV16 with this enzyme highlight the importance of selectively targeting furin as a potential antiviral therapeutic approach.

## 1. Introduction

The cleavage of viral surface proteins by the proprotein convertase (PC) family of proteases is an activating entry regulatory mechanism commonly associated with the virulence of some highly pathogenic viruses [1]. The localization of the PCs, furin, PC4, PC5, PACE4, and PC7 to the pericellular environment of the constitutive protein secretion pathway makes these proteases susceptible to exploitation by viruses during entry and egress [2]. In addition to some variations in their subcellular distribution, PCs also vary in reactivity. Therefore, virus PC reaction specificity and selectivity differences can potentially influence virus infectivity and tropism. HPV was used in this study as a model system to investigate variability in PC reactivity for virus activation.

Furin is the most reactive PC, and its substrate specificity differs from the other PCs because of differences in active site and exosite interactions with the substrate [3,4]. Natural inhibitors of PCs are serpins, and many organisms have serpin-based PC inhibitors [5,6]. Serpin B8 is a mammalian PC inhibitor that preferentially reacts with furin over the other PCs [3,7]. The serpin α1PDX is a PC inhibitor of high specificity and general selectivity derived by engineering PC reactive site specificity onto the human serpin α1-antitrypsin [3,8]. The homologous addition of serpin B8 reactive site and exosite residues to α1-antitrypsin generated the furin-selective inhibitor α1ORD (140–400-fold higher specificity) and the inhibitor α1MDW (20–60-fold higher specificity), which is selective for the other PCs [3].

Papillomaviruses have nonenveloped capsids composed of the two viral late expression proteins L1 and L2. Capsids consist of 360 L1 monomers arranged into 72 pentamers, or capsomers, and 20–72 L2 monomers distributed one per capsomer [9,10,11]. The cleavage at the residue L2-Arg12 in HPV16 occurs before or during entry [12]. This cleavage is essential for virus activation and entry, as demonstrated using in vitro synthesized HPV pseudoviruses (PsV) [13]. The PC motif at this site is universally conserved at the homologous position in all the HPV types. Papillomaviruses are double-stranded circular DNA viruses that proliferate in differentiating epithelial keratinocytes. The viral early expression proteins E6 and E7 interfere with the cell’s replication mechanism to favor a cellular environment of continuous growth propitious to viral particle production that may increase the cell propensity for oncogenic transformation [14,15].

There are about 200 HPV genotypes, or types, that infect cutaneous and mucosal epithelium [16]. HPV types are strongly associated with specific anatomic sites of infection [17]. The oncogenic mucosal HPV types 16 and 18 are associated with most HPV-related anogenital and oropharyngeal cancers [18,19,20]. HPV16 has the highest transmission, persistency, and transformation rates and is the most prevalent type, followed by HPV18, HPV52, HPV31, and HPV58 [21,22,23]. HPV16 is associated with over 60% of oropharyngeal cancers and HPV 31, 33, and 35 with the rest [24]. HPV18 rarely appears in the oropharyngeal cavity. HPV-related cancers occur primarily in the transformation zone of the cervix and the tonsillar crypts, with HPV16 having a higher association with squamous cell carcinoma, whereas HPV18 associates more often with adenocarcinomas. Therefore, there seems to be a correlation between anatomic site tropism and oncogenesis particular to each HPV type for which pre- and post-entry events may contribute to it [25,26,27,28,29].

Because the proteolytic processing of the capsid is a rate-limiting step for virus entry, any diversity of PC gene expression in keratinocytes at anatomic sites may selectively influence the entry of HPV types if viruses reacted differently with PCs. Keratinocytes from different common human anatomic sites of HPV infection express genes differently, such as those involved with immunity [30]. Our study determined that keratinocytes’ total PC gene expression narrowly varied among various anatomic sites. The pattern of gene expression of the five PC genes was also similar among sites, with furin being the most expressed PC. Also, PsVs of various mucosal HPV types of high, moderate, and low oncogenic risk reacted significantly different in specificity and selectivity with PCs. HPV16 reacted with PCs with the broadest selectivity and highest specificity and produced the highest entry values compared to the other HPV types.

## 2. Materials and Methods

### 2.1. Quantification of PC Gene Expression

The PC gene expression in primary keratinocytes was measured following the same methodology described previously [31]. Primary human keratinocytes were isolated from biopsy specimens from 2 to 7 donors per anatomic site and combined to form a pool of cells that were grown in organotypic raft cultures after the first or second passages. Raft tissues were harvested after 20 days. mRNA was isolated using RNeasy kit (Qiagen, Germantown, MD, USA), and cDNA synthesis was performed using the Maxima First Stranded kit (Thermo Fisher Scientific, Waltham, MA, USA). Quantification of transcriptional gene expression by RT-qPCR was conducted using the standard SYBR Select Master Mix (Applied Biosystems, Life Technologies, Waltham, MA, USA) and performed in a Bio-Rad CFX96 Real-Time System thermal cycler programmed for 40 cycles. The corresponding forward and reverse primers based on the human sequences for each gene were obtained from The Primer Bank and synthesized by Integrated DNA Technologies (Coralville, IA, USA, and those were, for furin (ID 20336193c2), 5′-tcggggactattaccacttctg-3′ and 5′-ccagccactgtacttgaggc-3′; for PC4 (ID 20336189c9), 5′-gctgccggtcggaaatgaa-3′ and 5′-gtcgtagctggcgtaggaat-3′; for PC5 (ID 207030317c2), 5′-gagggacccacagtttcatttc-3′ and 5′-tgggcacgactgaagtcataa-3′; for PACE4 (ID 20336189c2), 5′-gctgccggtcggaaatgaa-3′ and 5′-gtcgtagctggcgtaggaat; for PC7 (ID 20336247c1), 5′-gcagcgtccacttcaacga-3′ and 5′-gcccagtcacattgcgttc-3′; and, for GAPDH (ID 378404907c1), 5′-ggagcgagatccctccaaaat-3′ and 5′-ggctgttgtcatacttctcatgg-3′.

### 2.2. Synthesis of PsV Particles

The HPV pseudovirions were produced following established procedures [32,33,34]. The plasmids encoding the capsid proteins L1 and L2 were obtained originally from Dr. Schiller at the National Institutes of Health in the USA (p18SheLL and p45SheLL), Dr. Muller at the German Cancer Research Center (p6SheLL and p16SheLL), and Dr. Kanda at the National Institute of Infectious Diseases in Japan (p31SheLL, p52SheLL, and p58SheLL) and all kindly provided by Dr. Schiller. The plasmid pcDNA-GFP used as pseudogenome for reporting purposes and the cell line HEK293TT were also kindly provided by Dr. Schiller. Cells were grown adherent to collagen (Type I from rat tail, Corning, Corning, NY, USA) coated plates in high-glucose DMEM supplemented with 10% fetal bovine serum and 400 μg/mL Hygromycin B (Invitrogen, Carlsbad, CA, USA). PsV particles were extracted three days post-transfection and allowed to mature overnight, followed by DNAse I treatment and further purified by size exclusion filtration using Sepharose 2B. Elution of pseudovirion particles was monitored by DNA absorbance at 260 nm and of protein absorbance at 280 nm, and fractions were pooled together and concentrated by filtration (Amicon ultra 0.5 mL 10K filters, Millipore, Burlington, MA, USA). Pseudovirion stocks were stored at −80 °C in aliquots for further experimentation.

### 2.3. Production of Recombinant PCs

Enzymatically active PCs were produced as described previously [3,4]. Truncated forms of the enzymes containing the catalytic domain and the regulatory P-domain were produced in Sf9 insect cells using the baculovirus expression system (Invitrogen). The secreted enzymes were purified from culture supernatants by nickel-affinity chromatography, which was supported by a poly-histidine tail engineered at the carboxyl terminal end of the proteins, followed by size exclusion chromatography. Care was given to remove glycerol during filtration and concentration as it interferes with the activation of PsV and cell entry assays. The concentration of active proteases was determined by active-site titrations using the covalent-bound inhibitor dec-Arg-Val-Lys-Arg-chloromethyl ketone (CMK, Bachem, Torrance, CA, USA) and an activity assay that employs pyr-Arg-Thr-Lys-Arg-amido-methylcoumarin (Bachem, Torrance, CA, USA) as substrate.

### 2.4. Production of PC Inhibitors

The PC inhibitors α1PDX, α1ORD, and α1MDW were produced as inclusion bodies in E. coli BL21 DE3 cells by expressing their genes from a pET expression vector after the T7 promoter under IPTG regulation [4,35]. The inclusion bodies were dissolved with guanidine-HCl, and the protein was refolded under dialysis and isolated by two steps of ionic exchange chromatography [36].

### 2.5. HPV PsV Entry Assay

The entry of PsV-GFP particles into HEK293TT cells was measured by adapting the procedure described by Buck et al. [32,33,34]. Cells were grown in 6-well plates adherent to collagen (Type I from rat tail, Corning, Corning, NY, USA) in high-glucose DMEM supplemented with 10% fetal bovine serum and 400 μg/mL Hygromycin B. PsV stock solutions were appropriately diluted in order to produce GFP+ cell counts within a measurable range, as HPV types differed in their entry values. Volumes of diluted stocks between 50 and 100 μL were added to 70–80% confluent cells grown under 2 mL of culture media. After three days of incubation, the plate wells were fully confluent and GFP+ cells were counted using the Auto Imaging System of the Evos FL Auto microscope (Life Technologies, Walthan, MA, USA). At least 20 cell counts were obtained from each well with two or more repeats for each experimental condition. As mentioned above, PsV stocks were appropriately diluted to produce 50–1000 GFP+ cell counts per microscope field at a 4× magnification depending on the HPV type, which were reproducible between PsV batches. This sensitivity range was appropriate for PsV activation and entry inhibition experiments. For some experiments in which the prior proteolytic activation of PsVs was studied, the cells’ PC activity needed to be eliminated. The PC inhibitor CMK (1 μM) was added to HEK293TT cultures 1 h before starting entry assays. This inhibitor concentration was enough to reduce GFP+ cell counts of non-preactivated PsVs by over 95% with most HPV types.

### 2.6. HPV Activation Reactions with PCs

Preactivation reaction mixtures consisted of 100 μL reaction volumes and were composed of 10 μL of PsV stocks, appropriately diluted as described above, PC solutions to reach 5 or 40 nM, and completed with buffer solution of 100 mM Hepes and 1 mM calcium chloride at pH 7.4. All buffer and PC solutions were filter sterilized beforehand using 0.22 μm Costar Spin-X centrifuge tube filters Corning, Corning, NY, USA). The reactions were incubated at 37 °C for the specified length of time. Each reaction mixture (100 μL) was added to a single cell culture well of a 6-well plate containing 2 mL of cell culture media for downstream cell entry assays.

## 3. Results

### 3.1. PC Gene Expression in Human Keratinocytes at Anatomic Sites of HPV Infection

Expression of the furin, PC4, PC5, PACE4, and PC7 genes was measured in human keratinocytes from representative anatomic sites of HPV infection, such as the cervix, anus, gingiva, tonsils, and foreskin. Primary keratinocyte isolates included several individual donors. PC gene expression was assessed in keratinocytes growing in cell-differentiating raft cultures derived from primary cells (Appendix A). The total PC gene expression narrowly varied among anatomic sites within a four-fold margin (Figure 1A). The proportional expression of individual PC genes was consistent among anatomic sites (Figure 1B). Furin was the most abundantly expressed PC (40–50%). PC4 and PACE4 followed, each enzyme representing 14–27% of the total PC gene expression. PC7 constituted 5–16% and PC5 only 2–10%.

### 3.2. Production of HPV-Type PsV Particles and Entry Assays

The synthesis of HPV pseudovirus (PsV-GFP, green fluorescence protein) reporter particles for HPV types of high oncogenic risk (16 and 18), moderate risk (31, 45, 52, and 58), and low risk (6) was performed following methodologies for the expression and purification of HPV PsVs used in a well-established entry assay that employs HEK293TT cells [32,33,34]. PsV entry is dependent on the target cell PC activity. The total PC gene expression in HEK293TT cells was determined at 9.1% of the GAPDH gene expression, which falls within the keratinocyte expression range (Appendix A). Similarly to keratinocytes, PC4, furin, and PACE4 were also found to be the predominantly expressed PC genes in the HEK293TT cells, although PC4 was expressed more abundantly than furin in HEK293TT. Analysis of PsVs by nanoparticle tracking analysis (Nanosight 300, Malvern Panalytical, Salisbury, UK) determined particle concentrations from the area under the distribution peaks (Appendix A). Factoring of particle concentrations resulted in standardized entry values (Figure 2). HPV16 was the type that produced the highest entry values, followed by HPV45 and HPV58. The lowest entry values were those of HPV31, which were 21-fold lower than HPV16.

### 3.3. Reactivity of HPV Types with Furin

In order to test the reactivity of the HPV types with furin without the interference from the target cell PC activity, PsV particles were pretreated with 40 nM furin starting three hours prior to initiating entry assays into HEK293TT cells that had their PC activity inhibited beforehand with 1 μM of the general PC inhibitor CMK. Inhibiting the cell PC activity reduced entry values by over 95% for non-preactivated PsVs (compare Figure 2 versus black bars in Figure 3A). This means that 5% or less of the PsV particles were preactivated during synthesis; therefore, they produced baseline GFP+ cell counts resistant to inhibition by CMK. Only HPV45 retained a significantly higher baseline entry activity (~15%). A batch of HPV45 was produced in cells treated with 1 μM CMK. That treatment reduced the baseline entry to ~5%, suggesting that HPV45 is better activated during viral particle production compared to the other types. Preactivation with furin allowed the PsV particles to bypass the inhibition of the target cell PCs to gain entry (Figure 3A). Most HPV types produced higher entry values when preactivated with furin than when they were dependent on the target cell PCs for activation (Appendix A). HPV18 produced similar entry values when activated by the cell PC activity or furin. HPV6 had the largest entry enhancement (5.9-fold) upon furin pretreatment. Similar to the entry promoted by the cell PC activity (Figure 2), HPV16 also produced the highest entry values after preactivation with furin (Figure 3A). In order to further analyze the differences in reactivity of the HPV types with furin, PsVs were preactivated with furin in a time-dependent manner. HPV particles were incubated with 5 nM furin for different time points up to six hours, and incubations were followed by entry assays into CMK-treated HEK293TT cells. The kinetic curves showed significant differences in the rate of proteolytic preactivation among HPV types (Figure 3B). The faster kinetic curves were sigmoidal in shape (HPV16 and 58). All curves were fitted to the sigmoidal equation by linear regression analysis to estimate differences in reactivity based on the time of the inflection point (Appendix A). HPV16 was the most reactive type, with the shortest time for its inflection point (Appendix A). A 16-fold difference in reactivity between the fastest, HPV16, and the slowest, HPV18, was estimated.

### 3.4. Activation of HPV Types with PCs

In order to investigate the contribution of individual PCs to the activation of HPV types, PsV particles were pretreated with a PC at a concentration of 40 nM overnight, followed by entry assays into CMK-treated HEK293TT cells. Entry values were standardized to the entry value obtained with the PC-untreated PsVs into CMK-untreated cells, which was taken as 100% entry for each type (Figure 4, black control bars). There was considerable variability in the GFP+ cell counts produced by HPV types preactivated by the different PCs (Figure 4, gray bars). All PCs strongly preactivated HPV16 and HPV6, the latter to levels above the standard value. The preactivation of HPV18 was weak and dominated by furin, consistent with the furin preactivation experiments shown in Figure 3. Preactivation of HPV45 was also weak with all PCs, especially PC5. All PCs preactivated HPV31 except PC4. Several PCs, especially PC5, strongly preactivated HPV52. Furin strongly preactivated HPV58, followed by PC5 and PACE4.

Overnight incubations with PCs also produced entry values below the reference control with some HPV types in the absence of CMK (Figure 4, black bars). These results suggest that PCs may cleave the HPV capsid proteins at an additional site(s), negatively affecting virus entry, therefore, reducing GFP+ cell counts. This entry loss was observed in the reactions of HPV16 with furin, HPV18 with several PCs, HPV45 and HPV52 with PACE4, and HPV58 with several PCs. The preactivation of HPV31 or HPV6 produced no losses of entry.

In order to validate our approach using PsVs, the same treatment with PCs was repeated but using native HPV16, which led to increased infection of HaCat cells (Appendix A), and confirmed that PCs also preactivate native HPV16 for entry. Together, these results show considerable variability in PC reaction selectivity among HPV types, with HPV16 being the type with the broadest PC selectivity as all the PCs strongly activated it.

### 3.5. Reduction in HPV Entry by PC-Selective Inhibitors

Another approach employed to analyze PC selectivity in the activation of HPV types was targeting the cells’ PC activity with PC-selective inhibitors. The feasibility of using an exogenous protein-based inhibitor (α1PDX) compared to a small molecule inhibitor (CMK) was first verified. These two PC inhibitors are nonselective but highly specific. They strongly inhibited the entry of HPV types at low nanomolar concentrations when added to HEK293TT cells one hour before starting entry assays (Figure 5). The cell viability at a concentration of 500 nM CMK or α1PDX was not different from the untreated cells (Appendix A). Entry assays were started at a cell confluence of about 75%, and GFP+ cells were counted at full confluence three days later in all cases; therefore, the inhibitors did not affect cell growth. At 400 nM inhibitor concentration, the furin-selective inhibitor, α1ORD, reduced the entry of HPV16 by 55%. In contrast, α1MDW, an inhibitor selective for the non-furin PCs, reduced entry by 15%. These results suggest that furin contributes to HPV16 activation more than the other PCs in HEK293TT cells. The sharper decrease in HPV16 entry caused by α1PDX and CMK, compared to the selective inhibitors, likely reflects the high specificity of both inhibitors for all PCs [3]. α1ORD inhibited the entry of HPV18 by 53% but α1MDW by 8%. HPV types 31 and 52 were also inhibited more effectively by α1ORD than by α1MDW. HPV31 was more susceptible to inhibition than the other types. These studies further show the intrinsic variability in the activation of HPV types by the different PCs. Also, they show the effectiveness of PC-selective inhibitors in reducing the entry of a virus by inhibiting PCs in a cellular assay context.

## 4. Discussion

This comparative study used recombinant PCs, selective PC inhibitors, and pseudovirions to describe variability in the proteolytic activation of HPV types. It showed differences in PC specificity and selectivity among virus types. HPV16, the most prevalent HPV type and the one associated with most anogenital and oropharyngeal HPV-related cancers [21,22,23,24], had the highest entry values and reacted with PCs with the highest specificity and broadest selectivity.

The expression of the furin, PC4, PC5, PACE4, and PC7 genes was measured in primary human keratinocytes from several representative anatomic sites of HPV infection, such as the cervix, anus, tonsils, gingiva, and foreskin. Total PC gene expression varied narrowly within a 4-fold factor range among sites, and the pattern of individual PC gene expression was similar. Furin was the predominant PC, and PC5 was the least expressed gene. Furin, PC4, and PACE4 were the PCs with the highest gene expression levels and, therefore, the most likely to contribute to HPV activation at anatomic sites in vivo. Currently, there are no reliable antibodies specific for each of the PCs; neither are there substrates that could distinguish activity from any specific PC; therefore, the protein quantification of individual PCs was not possible.

HPV16 had the highest PsV concentration-standardized entry values promoted by the PC activity of target cells or recombinant furin in pretreatment reactions. However, entry values were higher when entry was promoted by the furin pretreatment for the HPV types 6, 16, 45, 52, and 58. HPV18 and HPV31 had the lowest entry values, and their activation was not or was barely enhanced by the furin pretreatment, which indicates a limitation to activation determined by an unknown factor. These observations are consistent with previously reported differences in furin dependence between HPV16 and HPV18. HPV16 can be efficiently preactivated by furin without additional factors, as shown here and by others [13,37], while HPV18 may need the involvement of HSPGs [28,37]. Other cell factors may also influence the activation of HPV types. The entry of HPV6 after preactivation with all PCs exceeded the entry promoted by the cell PC activity, which means that, in contrast to HPV18, activation of HPV6 on the cell surface was disfavored compared to the direct preactivation with PCs. Similar to HPV18, the preactivation of HPV45 was also inefficient. These results align with other studies that suggest that differences in the binding to polysaccharides may influence pre-entry events contributing to differences in the preference of anatomic site amongst high-risk HPV types [29]. In conclusion, the interaction of the virus with the host cells may promote or hinder proteolytic activation, depending on the virus type.

The kinetic analysis of preactivation of HPV types with a constant concentration of furin showed that HPV16 had the fastest rate of preactivation. The sigmoidal pattern of the rate’s time dependence indicates that the proteolytic processing progressively increased with time. This pattern is characteristic of co-operative conformational changes, which may propagate across the capsid triggered by proteolytic cleavages. Differences in reactivity among HPV types may result from variability at the cleavage site amino acid sequences [1] or from differences in capsid conformation. Also, co-operativity may be affected by the capsid rigidity due to different levels of capsid maturation. Because the preactivation reactions were performed independently of cells and the effect of other influences on entry would be diluted during the long time between entry and the GFP+ cell counts, the sigmoidal rate pattern should be the result of the preactivation reactions and not of an asynchronous mode of virus internalization that was assumed to be associated with cellular determinants of virus entry [38]. Directly monitoring the cleavage of L2 and analyzing the capsid structure in several HPV types will be required to determine the origin of the rate differences. These studies were performed with pseudovirions, which may not necessarily behave exactly the same as the native viruses. Especially concerning is any difference in capsid maturation that may affect reactivity with the PCs. Still, the observed variability in PC reactivity among pseudovirion types should be representative of variability among actual viruses.

PC selectivity in the activation of HPV types was investigated by performing preactivation reactions with PCs and by treating the cells with PC-selective inhibitors. Both approaches demonstrated variations in PC selectivity among virus types. All PCs strongly preactivated HPV16. In contrast, HPV18 was preactivated mainly by furin. Other virus types showed a mix of preferences.

In previous biochemical studies, we reported the development of PC-selective serpin-based inhibitors [3]. Here, the inhibitors were used in cell-based assays of HPV entry. The two inhibitors demonstrated differential inhibition of several HPV types at concentrations below 1 μM, in which they react selectively. At higher concentrations, both inhibitors will increasingly react with all PCs.

Papillomaviruses show strict tropism to infection sites and are probably adapted to generate productive infections only at specifically localized keratinocytes. Entering the wrong cells may lead to nonproductive infections, induction of genome instability, and progression to malignancy [17]. The highest prevalence and cancer risk of HPV16 correlate with this virus having the highest PC reactivity and entry rates. Furin being the most expressed PC and the one HPV16 depends on the most for activation highlights the feasibility of selectively inhibiting furin as a potentially safer therapeutic approach than inhibiting all the PCs with nonselective inhibitors.

## 5. Conclusions

Microbes’ adaptation to the host conditions may result in some of them becoming highly successful at taking advantage of the host. HPV16 is highly infective and transmissible, and its efficient reactivity with the host PCs plays an essential role in its success. The presence of furin cleavage sites in viral envelope proteins indicates the potential for high virulence in some viruses. However, this potential depends on these cleavage sites efficiently reacting not only with furin but also with the other PCs.

## Figures and Tables

**Figure 1 viruses-16-00039-f001:**
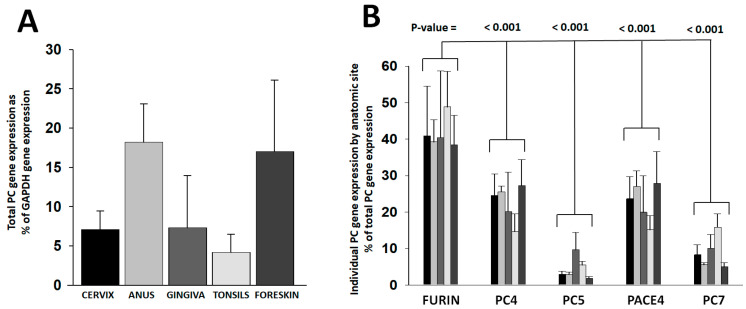
*PC gene expression in keratinocytes from anatomic sites of HPV infection.* PC gene expression was determined in keratinocytes grown in raft cultures out of primary cells isolated from different anatomic sites. Raw gene amplification values were standardized to the cDNA concentration (Appendix A). Plotted values are (**A**) the total PC gene expression values relative to the GAPDH gene expression and (**B**) the proportional expression of each PC gene compared to the total PC gene expressed per anatomic site (bar gray tonality code is as in panel (**A**)). Statistical difference significance *p*-values are shown for the comparison of the furin gene expression combined from all anatomic sites versus each one of the other PCs using two-tailed Student *t*-test.

**Figure 2 viruses-16-00039-f002:**
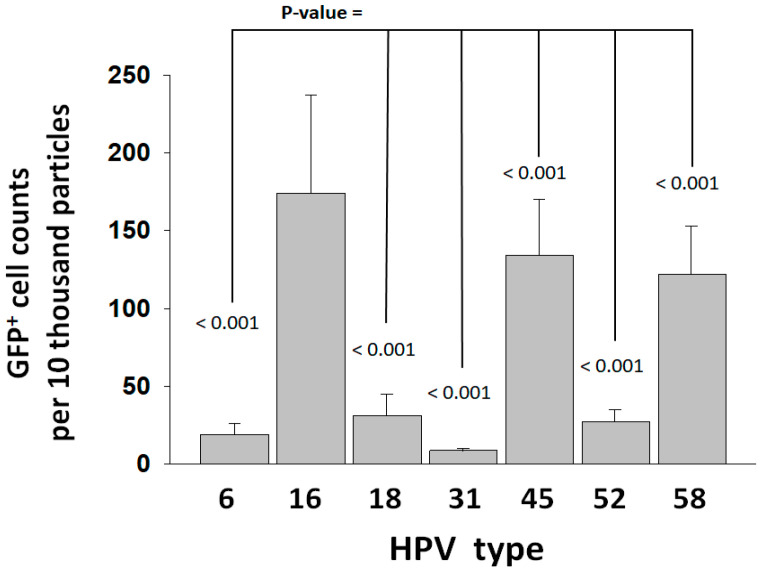
*Entry of HPV types into HEK293TT cells.* Entry was measured as GFP^+^ cell counts and standardized to PsV particle concentration (Appendix A). Entry values represent the average and standard deviation from two experiments consisting of 20 measurements each. Statistical difference significance *p*-values are shown for the comparison of the HPV16 GFP+ cell counts against each one of the other HPV types using two-tailed Student *t*-test.

**Figure 3 viruses-16-00039-f003:**
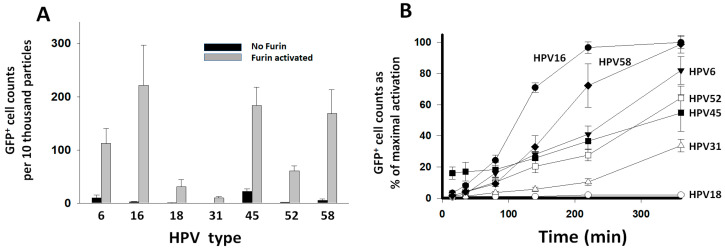
*Preactivation of HPV types with furin.* PsV were preactivated with furin and entry was measured in HEK293TT cells that had their PC activity inhibited with 1 μM CMK. Entry was measured as GFP+ cell counts and standardized to PsV particle concentration (Appendix A). Entry values represent the average and standard deviation from two experiments consisting of 20 measurements each. (**A**) PsVs were added to the cells without furin treatment (black bars) or after preactivation with 40 nM furin for 3 h (gray bars). (**B**). Time course reactions of PsV preactivation with 5 nM furin. Data points correspond to the proportional activation with respect to the entry values obtained after activation with furin in panel (**A**).

**Figure 4 viruses-16-00039-f004:**
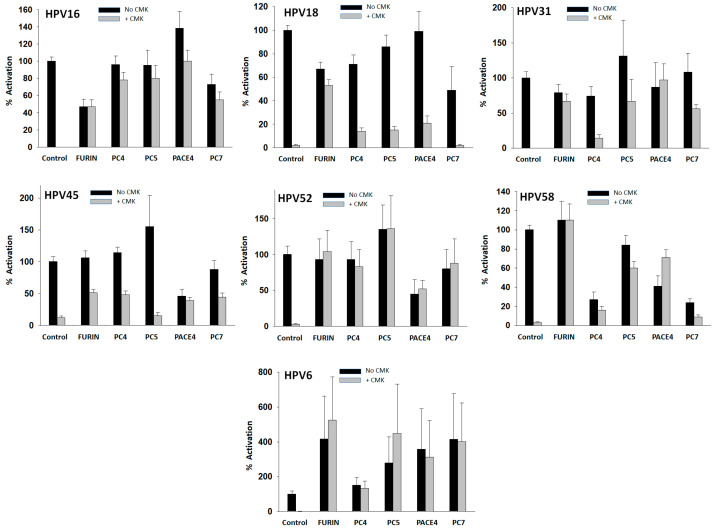
*Preactivation of HPV types with PCs.* PsV particles were pretreated with 40 nM of a PC overnight. Preactivation was measured by cell entry assays into CMK-treated HEK293TT cells (gray bars). Control (gray bars) values are the same as in Figure 3A without furin. All entry values are plotted as activation relative to the entry produced by PC-untreated PsV particles into CMK-untreated cells (shown also in Figure 2) that was taken as the 100% value (control, black bars). Data values represent the average and standard deviation from at least two experiments with 20 measurements in each one.

**Figure 5 viruses-16-00039-f005:**
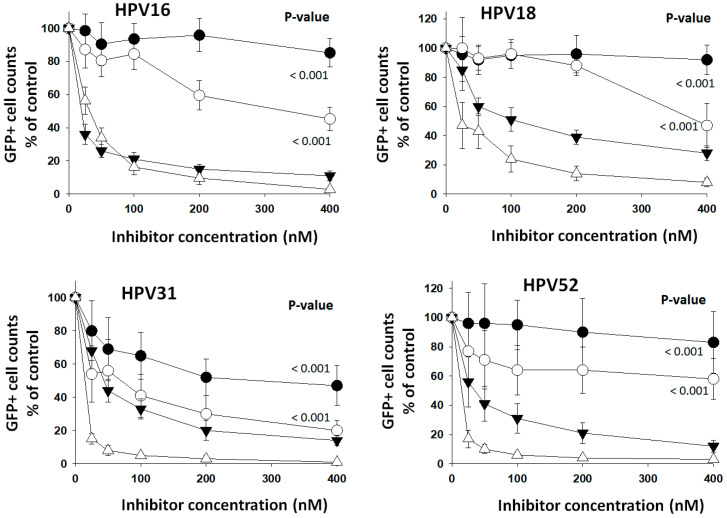
*Selective PC inhibitors reduced entry of HPV types.* Increasing concentrations of the selective PC inhibitors α1ORD (white circles) and α1MDW (black circles) and of the general PC inhibitors α1PDX (black triangles) and CMK (white triangles) were added to HEK293TT cells one hour ahead of starting entry assays with HPV types. Data points represent the average and standard deviation from two experiments with 20 measurements each. Statistical difference significance *p*-values are shown for the effect of 400 nM α1ORD and α1MDW on HPV-type GFP+ cell counts using two-tailed Student *t*-test.

## Data Availability

Data are contained within the article and Appendix A.

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
