# Peer review of "Diversity in Proprotein Convertase Reactivity among Human Papillomavirus Types"

_viruses, 2023, doi:10.3390/v16010039_

Round 1
Reviewer 1 Report
Comments and Suggestions for Authors
Please see comments to authors. There is more work that needs to be done, especially points 1,3,4,5.
In this manuscript, Izaguirre and colleagues continue their analysis of the role of proprotein convertases (PC) in cell entry by human papillomaviruses. The PC furin “activates” HPV for infection by cleaving the N-terminus of the L2 capsid protein. Here they compare the effect of different PCs on different HPV pseudovirus (PSV) types. Their overall conclusion is that HPV16, the most common high-risk HPV type, displays the highest entry values of the types tested and is activated by the broadest array of PCs, with a preference for furin. They also show that highly selective PC inhibitors can inhibit infection, although not as completely as broadly acting inhibitors. There are some interesting findings reported here, but also issues that reduce enthusiasm.
Major comments:
1. There is a major problem with Figure 1/Table S1. It is hard to reconcile the data in Fig 1A, with highest expression of total PC in anus and foreskin, with the individual expression shown in Fig 1B, where, for example, cervix and anus are very similar for all genes examined. In fact, table S1 does not correlate with Fig 1B because, for example, in the table anus is higher than cervix for furin and PC4, which is not what is shown in the figure.
2. Although they show differences in mRNA expression, how does that correlate with relative protein expression or activity?
3. They purify pseudovirus by size exclusion chromatography, which will not separate capsids containing the reporter plasmid from empty capsids, which will affect measurement of entry activity. In the methods, they say they infected cells with “appropriately diluted” PSV stock solutions. What does this mean? Diluted according to particle numbers, or packaged reporter plasmids? If they diluted according to number of particles and different HPV types are more or less efficient at DNA packaging, this would affect their entry assays. Most groups purify by density gradient centrifugation to remove empty capsids.
4. The entry assays are poorly described in the results or figure/table legends. What cell types are used for entry, how long after infection, what was measured? Some of this information in is the methods but it should be clearer in the results. “GFP+ counts” seems a confusing way of saying GFP+ cells, which is what they are measuring.
5. Another major issue that is not considered is the extent of “precleavage” in the various virus preparations. Different virus types may have different amounts of cleavage during production due to sequence differences in L2, which will influence the results and interpretation. For example. If HPV18 is mostly cleaved before it is added to cells, then addition of exogenous PCs would be expected to have minimal effect.
6. Related to point 4, this work would be more compelling if they actually measured cleavage of L2 from different HPV types with different PCs.
Minor points
7. It is better to refer to L1 and L2 as capsid proteins, not coat proteins, since coat proteins more often is used in relation to enveloped viruses.
8. Line 52, L1 is not “divided” into 72 capsomers. Perhaps organized into capsomers or exists in capsomers.
9. Last section of the results should be headed “PC inhibitors inhibit HPV entry”, not block entry, because most don’t block it.
Author Response
- There is a major problem with Figure 1/Table S1. It is hard to reconcile the data in Fig 1A, with highest expression of total PC in anus and foreskin, with the individual expression shown in Fig 1B, where, for example, cervix and anus are very similar for all genes examined. In fact, table S1 does not correlate with Fig 1B because, for example, in the table anus is higher than cervix for furin and PC4, which is not what is shown in the figure. ……… The data in Table S1 is just the raw gene expression data standardized to cDNA concentration. The plots in Figure 1B represent percentages of gene expression relative to GAPDH gene expression and to the total PC gene expression per anatomic site. Legends to Figure 1 and Table S1 have been expanded to clarify the presentation of the data.
- Although they show differences in mRNA expression, how does that correlate with relative protein expression or activity? ……….A couple of sentences have been added in the discussion to address the lack of protein expression due to the lack of reliable specific antibodies and substrates that would allow performing estimation of protein expression or activity.
- They purify pseudovirus by size exclusion chromatography, which will not separate capsids containing the reporter plasmid from empty capsids, which will affect measurement of entry activity. ……….As described by the Buck group, the fraction of particles containing the reported plasmid is actually less than 1%. All other particles are empty or contain random pieces of genomic DNA. The entry assay is well represented by the small fraction of “hot” particles loaded with the reporter. In the methods, they say they infected cells with “appropriately diluted” PSV stock solutions. What does this mean? Diluted according to particle numbers, or packaged reporter plasmids? If they diluted according to number of particles and different HPV types are more or less efficient at DNA packaging, this would affect their entry assays. Most groups purify by density gradient centrifugation to remove empty capsids. ………The “appropriate diluted” expression has been removed and substituted with a better explanation that clarifies that HPV type stocks were diluted to generate between 50 to 1000 GFP+ cell counts per microscope field to allow measurable treatments as each type generates different counts per particle concentration.
- The entry assays are poorly described in the results or figure/table legends. What cell types are used for entry, how long after infection, what was measured? Some of this information in is the methods but it should be clearer in the results. “GFP+ counts” seems a confusing way of saying GFP+ cells, which is what they are measuring. ………The description of the entry assay methodology has been expanded in the Methods section as well as in the figure legends. The GFP+ counts” expression has been substituted with “GFP+ cell counts” all throughout the manuscript and figures.
- Another major issue that is not considered is the extent of “precleavage” in the various virus preparations. Different virus types may have different amounts of cleavage during production due to sequence differences in L2, which will influence the results and interpretation. For example. If HPV18 is mostly cleaved before it is added to cells, then addition of exogenous PCs would be expected to have minimal effect. ……….Figure 3A shows the baseline entry of all HPV types studied into HEK293TT cells treated with CMK, which means that only PsVs cleaved during production are entry capable. Most types had a baseline of 5% or less, only the HPV45 baseline reached 15%. Those baseline levels did not interfere with the interpretation of the experiments.
- Related to point 4, this work would be more compelling if they actually measured cleavage of L2 from different HPV types with different PCs……….. Measuring L2 cleavage requires adding epitope tags to the protein. We tried, but the low quality of the measurements and the effort of adding tags to several HPV types made the cost/gain ratio not favorable. This approach would be more effective in structural studies for the determination of the cooperative mechanism of the L2 cleavage.
Minor points
- It is better to refer to L1 and L2 as capsid proteins, not coat proteins, since coat proteins more often is used in relation to enveloped viruses. …….The word “coat” has been removed throughout and substituted by “capsid”.
- Line 52, L1 is not “divided” into 72 capsomers. Perhaps organized into capsomers or exists in capsomers. ………The sentence has been upgraded, and the word “divided” substituted by “arranged”.
- Last section of the results should be headed “PC inhibitors inhibit HPV entry”, not block entry, because most don’t block it. ………The head of the section is now “Reduction in HPV entry by PC-selective inhibitors”.
Reviewer 2 Report
Comments and Suggestions for Authors
SUMMARY and COMMENTS
In this work, with the aim of investigating the HPV cell entry mechanisms, the authors evaluated the variability in reactivity with furin (which allow the cleavage of viral surface proteins) and other proprotein convertases among several HPV types. The authors employed an in vitro model of primary human keratinocytes and HEK293TT cells for the experiments. Main data indicate that HPV16 reacted with furin at the highest rate compared to the other HPV types in time-dependent preactivation reactions and produced the highest entry values.
This work has potential, but several necessary improvements should be made. Here some comment/observation:
1. My only major concern is that statistical analysis are lacking and they should be performed when appropriate. Although numerous experiments/analyses were performed, the study is lacking in statistical analyses. In my opinion, without an adequate statistical analysis the work cannot be accepted for publication
2. Lines 58-59 in humans? Or in general?
3. Inlines 51-55 for completeness of information, I suggest including some brief information on the HPV genome, and its encoded proteins, including E proteins and their implication in HPV-driven carcinogenesis (doi: 10.1007/s10147-023-02337-7 and DOI: 10.1016/j.tvr.2023.200258)
4. Lines 61-64 these two additional references on HPV16 and other HPV typers should be included (DOI: 10.3390/pathogens9030224 and doi: 10.1371/journal.pone.0099114)
5. Red and blue colors throughout the text should be avoided. Moreover the citation style should be revised, as the journal viruses MDPI uses square parentheses and numbers
6. The primer sequences retrieved from “The Primer Bank” were from some study? If yes the studies should be cited
7. Methods, more information should be included on the establishment of primary human keratinocytes. Tissue source should be detailed here (methods). Isolation protocol? Have epithelial markers been evaluated? The same observation can be made on the RNA isolation method
8. Supplementary material should be removed from the main text. Concerning supplementary material, the quality of figure S1 is poor and should be improved.
9. In page 12, the figure is overlapping the figure caption, please fixt this problem
10. The readabi.ity of some figure should be improved. I suggest including figure legends of fig 3B, fig 5 and
11. The subhead title “conclusions” should be replaced with “discussion”
12. For a better reading tables/figures should not be mentioned in the discussion
13. The main limitations of the study should be included
14. A brief conclusion should be included at the end of the discussion
Author Response
- My only major concern is that statistical analysis are lacking and they should be performed when appropriate. Although numerous experiments/analyses were performed, the study is lacking in statistical analyses. In my opinion, without an adequate statistical analysis the work cannot be accepted for publication. ………Student t-tests have been performed to experiments presented in Figure 1B, Figure 2, and Figure 5. P-values are now inserted into the figures.
- Lines 58-59 in humans? Or in general? Humans
- Inlines 51-55 for completeness of information, I suggest including some brief information on the HPV genome, and its encoded proteins, including E proteins and their implication in HPV-driven carcinogenesis (doi: 10.1007/s10147-023-02337-7 and DOI: 10.1016/j.tvr.2023.200258). ……..The following sentence with the references was added: “Papillomaviruses are double stranded circular DNA viruses that proliferate in differentiating epithelial keratinocytes. The viral early expression proteins E6 and E7 interfere with the cell’s replication mechanism to favor a cellular environment of continuous growth propitious to viral particle production that may increase the cell propensity for oncogenic transformation [14, 15].”
- Lines 61-64 these two additional references on HPV16 and other HPV typers should be included (DOI: 10.3390/pathogens9030224 and doi: 10.1371/journal.pone.0099114). ……..The two references were added.
- Red and blue colors throughout the text should be avoided. Moreover the citation style should be revised, as the journal viruses MDPI uses square parentheses and numbers. ………Colors were used only to help during writing and reviewing the manuscript.
- The primer sequences retrieved from “The Primer Bank” were from some study? If yes the studies should be cited. ……….The Bank does not cite studies but the ID numbers for each primer pair was added.
- Methods, more information should be included on the establishment of primary human keratinocytes. Tissue source should be detailed here (methods). Isolation protocol? Have epithelial markers been evaluated? The same observation can be made on the RNA isolation method. ………The tissue sample isolation was described in a previous publication, which is referenced. More information was added to provide more clarity.
- Supplementary material should be removed from the main text. Concerning supplementary material, the quality of figure S1 is poor and should be improved. ………..All supplementary material has been separated into a supplement section. All figures in the originally submitted manuscript were preliminary images generated in Power Point. All figures in the final manuscript and supplement have been upgraded to high resolution images generated in Adobe Photoshop.
- In page 12, the figure is overlapping the figure caption, please fixt this problem. …….Done
- The readabi.ity of some figure should be improved. I suggest including figure legends of fig 3B, fig 5 and. ……….Images and legends have been improved.
- The subhead title “conclusions” should be replaced with “discussion”. ………A Discussion section is now followed by a short Conclusion.
- For a better reading tables/figures should not be mentioned in the discussion. ……….All references to figures and tables in the Discussion section have been removed.
- The main limitations of the study should be included. ……….The following statement was added to the Discussion “These studies were performed with pseudovirions, which may not necessarily behave exactly the same as the native viruses. Especially concerning is any difference in capsid maturation that may affect reactivity with the PCs. Still, the observed variability in PC reactivity among pseudovirion types should be representative of variability among actual viruses.”
- A brief conclusion should be included at the end of the discussion. ……..A Conclusion section was added.
Reviewer 3 Report
Comments and Suggestions for Authors
This manuscript broadly explores: “The cleavage of viral surface proteins by furin is associated with some viruses’ high virulence and infectivity. The human papillomavirus (HPV) requires the proteolytic processing of its coat proteins for activation before entry. Variability in reactivity with furin and other proprotein convertases (PCs) among HPV types was investigated.” The tissue distribution of the various PCs was measured, with furin found to be dominant. The entry of HPV types was studied with and without furin preactivation using a model system, and little entry was observed in the absence of that preactivation. HPV16 showed the fastest cell entry, with HPV58 next fastest and HPV18 the slowest of the seven types examined.
Line 60-61. To my understanding, HPV is associated with many oropharyngeal cancers but has not been proven as the causative agent in the way that HPV16 and 18 cause 70% of HPV-derived cervical cancers. Here is a quote from reference 19, “The human papillomavirus (HPV) is an etiologic agent associated with the development of head and neck squamous carcinoma (HNSCC)—in particular, oropharyngeal squamous cell carcinoma.” Please note the use of the term ‘associated.’ This is not merely a pedantic point.
267-8 Figure S3 overlays the text and what may be Figure 4.
294-295. Same problem as lines 60-61.
Author Response
Line 60-61. To my understanding, HPV is associated with many oropharyngeal cancers but has not been proven as the causative agent in the way that HPV16 and 18 cause 70% of HPV-derived cervical cancers. Here is a quote from reference 19, “The human papillomavirus (HPV) is an etiologic agent associated with the development of head and neck squamous carcinoma (HNSCC)—in particular, oropharyngeal squamous cell carcinoma.” Please note the use of the term ‘associated.’ This is not merely a pedantic point. ……..The word “associated” is now used as a more cautionary statement.
267-8 Figure S3 overlays the text and what may be Figure 4. ………The problem has been fixed.
294-295. Same problem as lines 60-61. ………Resolved.
Round 2
Reviewer 1 Report
Comments and Suggestions for Authors
The authors responded adequately to my comments.